# Structure of voltage-modulated sodium-selective NALCN-FAM155A channel complex

Yunlu Kang [1], Jing-Xiang Wu[1,2,3] & Lei Chen [1,2,3 ✉]

Resting membrane potential determines the excitability of the cell and is essential for the cellular electrical activities. The NALCN channel mediates sodium leak currents, which positively adjust resting membrane potential towards depolarization. The NALCN channel is involved in several neurological processes and has been implicated in a spectrum of neurodevelopmental diseases. Here, we report the cryo-EM structure of rat NALCN and mouse FAM155A complex to 2.7 Å resolution. The structure reveals detailed interactions between NALCN and the extracellular cysteine-rich domain of FAM155A. We find that the non-canonical architecture of NALCN selectivity filter dictates its sodium selectivity and calcium block, and that the asymmetric arrangement of two functional voltage sensors confers the modulation by membrane potential. Moreover, mutations associated with human diseases map to the domain-domain interfaces or the pore domain of NALCN, intuitively suggesting their pathological mechanisms.

---

[1] State Key Laboratory of Membrane Biology, College of Future Technology, Institute of Molecular Medicine, Beijing Key Laboratory of Cardiometabolic Molecular Medicine, Peking University, 100871 Beijing, China. [2] Peking-Tsinghua Center for Life Sciences, Peking University, 100871 Beijing, China. [3] Academy for Advanced Interdisciplinary Studies, Peking University, 100871 Beijing, China. ✉email: chenlei2016@pku.edu.cn

NALCN channel is a voltage-modulated sodium-selective ion channel[1]. It mediates sodium leak currents that are essential for setting the membrane excitability[2]. NALCN channel is involved in several crucial physiological processes including maintaining normal circadian rhythms[3] and respiratory rhythm[4]. NALCN channel functions as a multi-protein complex, which consists of NALCN, FAM155, UNC79, UNC80, and other proteins[5]. Mutations in the genes of human NALCN or UNC80 lead to a spectrum of neurological diseases[6], including infantile hypotonia, psychomotor retardation, and characteristic facies (IHPRF)[7,8] and congenital contractures of the limbs and face, hypotonia, and developmental delay (CLIFAHDD)[9]. In addition, a gain-of-function mutation of NALCN in the mouse causes the *dreamless* phenotype, suggesting the function of NALCN in rapid eye movement sleep[10].

Sequence analysis shows NALCN protein shares a similar domain arrangement to the pore subunit of eukaryotic voltage-gated sodium channels $Na_V$ or calcium channels $Ca_V$. Four consecutive voltage sensor-pore modules are fused into a single polypeptide chain. However, unique to the NALCN channel, the positive charges on voltage sensors of NALCN are much fewer than that of $Na_V$ or $Ca_V$. Moreover, residues in the selectivity filter of NALCN are "EEKE" instead of "DEKA" of $Na_V$ or "EEEE" of $Ca_V$. These features render NALCN to be a special clade in the voltage-gated channel superfamily[11]. In addition to the NALCN subunit, FAM155, UNC79, and UNC80 subunits are indispensable for the function of NALCN channel. The co-expression of NALCN, FAM155, UNC79, and UNC80 proteins in a heterologous expression system, such as *Xenopus laevis* oocytes or HEK293 cells, is necessary and sufficient to generate robust voltage-modulated sodium-selective currents[1]. FAM155 is a transmembrane protein family with a cysteine-rich domain (CRD). There are two FAM155 family members in human, namely, FAM155A and FAM155B, both of which could support the currents of the NALCN complex[1]. FAM155 is the homolog of NLF-1 in *C. elegans*, which is required for the membrane localization of NALCN[12]. It is proposed that the interaction between NALCN and FAM155 is similar to the interaction between yeast calcium channel Cch1 and its regulator Mid1[13,14]. The other two proteins in the complex, UNC79 and UNC80, are large conserved proteins without any known domains but they are essential for the function of NALCN in vivo[15–17]. During the preparation of this manuscript, the structure of human NALCN–FAM155A at 2.8 Å in nanodiscs was published[18] and another group reported a similar structure at 3.1 Å in detergent on preprint sever[19]. Here, we describe the structure of rat NALCN and mouse FAM155A in detergent to the resolution of 2.7 Å, which provides enhanced details and additional structural insights into the mechanism of this important channel complex.

## Results

**Structure of the NALCN–FAM155A core complex**. Despite extensive screening of NALCN homologs and optimization, we failed to purify the stable hetero-tetrameric complex of NALCN–FAM155–UNC79–UNC80. However, we found C-terminal GFP tagged rat NALCN and C-terminal flag tagged mouse FAM155A can generate voltage-modulated NALCN currents upon co-expression of mouse UNC79 and mouse UNC80 (Supplementary Fig. 1a). Moreover, rat NALCN and mouse FAM155A can form a stable core complex which can be purified chromatographically (Supplementary Fig. 1b–d). Because NALCN protein dictates the properties of ion permeation, divalent ion block and voltage modulation[1] and FAM155 are required for correct trafficking of NALCN[12], we reasoned the structure of NALCN–FAM155A complex would provide clues of how NALCN channel works.

Therefore, we started with the cryo-EM studies of the NALCN–FAM155 core complex. We prepared the NALCN–FAM155A complex in a buffer with sodium chloride and GDN detergent, mimicking the symmetric sodium solution used in electrophysiology recording (Supplementary Fig. 1a). The single-particle cryo-EM analysis yielded a map at a nominal resolution of 2.7 Å (Supplementary Figs. 2–4, and Supplementary Table 1). The map quality was sufficient for modeling the stable portions of NALCN and FAM155A which encompass 73% of NALCN and 37% of FAM155A. The excellent map quality also reveals several putative detergents or co-purified lipid densities (Supplementary Fig. 4b).

NALCN subunit shows a typical eukaryotic $Na_V$ or $Ca_V$ structure (Fig. 1a–d). Pore domains of domain I, II, III, and IV shape the ion permeation pathway (Fig. 1a–d). Voltage sensors (VS, S1–S4 segments) pack around the pore in a domain-swapped fashion (Fig. 1a–d). Interestingly, the arrangements of VS are highly asymmetric, with the voltage sensor of domain I (DI-VS) being the most off-axis (Fig. 1d). On the intracellular side, the C-terminal domain of NALCN binds onto the intracellular III–IV linker. The majority of the long II–III loop and the C-terminal 167 residues of NALCN were disordered. On the extracellular side, the CRD of FAM155A protrudes from the top of the complex into the extracellular space while the two predicted transmembrane helices of FAM155A were invisible, probably due to their flexibility. The structure of NALCN allowed us to locate all of the reported missense mutations identified in the NALCN subunit that cause diseases in human and model organisms (Fig. 1e, f)[7,9,10,17,20–27]. The structure presented here is overall similar to the recently published structure of human NALCN–FAM155A complex (EMD-22203 and PDB ID: 6XIW)[18] (Supplementary Fig. 7a), with a root-mean-square deviation of 0.68 Å. However, they differ in ion selectivity filter and cytosolic domain, as described later.

**Structure of FAM155A CRD and its interaction with NALCN**. The FAM155A has an extracellular cysteine-rich domain (CRD) which is folded by three consecutive α helices and the connecting loops in between. The CRD of FAM155A is highly cross-linked by disulfide bonds owing to the high content of cysteines in this domain. Most of the Cys residue pairs are conserved from NLF-1 of *C. elegans* to FAM155A/B in human, suggesting their important functions, presumably by stabilizing the structure of CRD (Fig. 2a–c and Supplementary Fig. 6).

The CRD of FAM155A binds at the top of the NALCN channel (Fig. 1b, c). Extracellular loops of domain I, III, and IV ($L5_I$, $L5_{III}$, $L5_{IV}$, and $L6_{IV}$) in NALCN form a platform for FAM155A binding (Fig. 2d). The post-α3 loop of FAM155A extends over the extracellular ion entrance of the NALCN pore (Fig. 2e). The $L5_{III}$ of NALCN covers above the post-α3 loop of FAM155A and locks it in the sandwiched conformation (Fig. 2e). The binding of CRD of FAM155A onto NALCN relies on its extensive interactions with $L5_{III}$, $L5_{IV}$, and $L6_{IV}$ of NALCN. In detail, K287 and E290 on α1 of FAM155A make cation-π and electrostatic interactions with W1085 and R1062 on $L5_{III}$, respectively (Fig. 2d). K307 on α1-α2 loop interacts with D1408 on $L6_{IV}$ (Fig. 2d). R1094 on $L5_{III}$ interacts with the carbonyl group of G364 of FAM155A, D1397 on $P2_{IV}$, and E1364 on $L5_{IV}$ (Fig. 2e). These interacting residues are highly conserved (Fig. 2b, c and Supplementary Figs. 5, 6). Notably, it is reported that mutation of R1094Q, which locates on the interface between NALCN and FAM155A (Fig. 2e), leads to the disordered respiratory rhythm with central apnea[27], suggesting the importance of these inter-subunit interactions.

**The pore domain of NALCN**. Along the ion permeation pathway, negatively charged residues are broadly distributed. They may function to attract sodium ions for permeation (Fig. 3a).

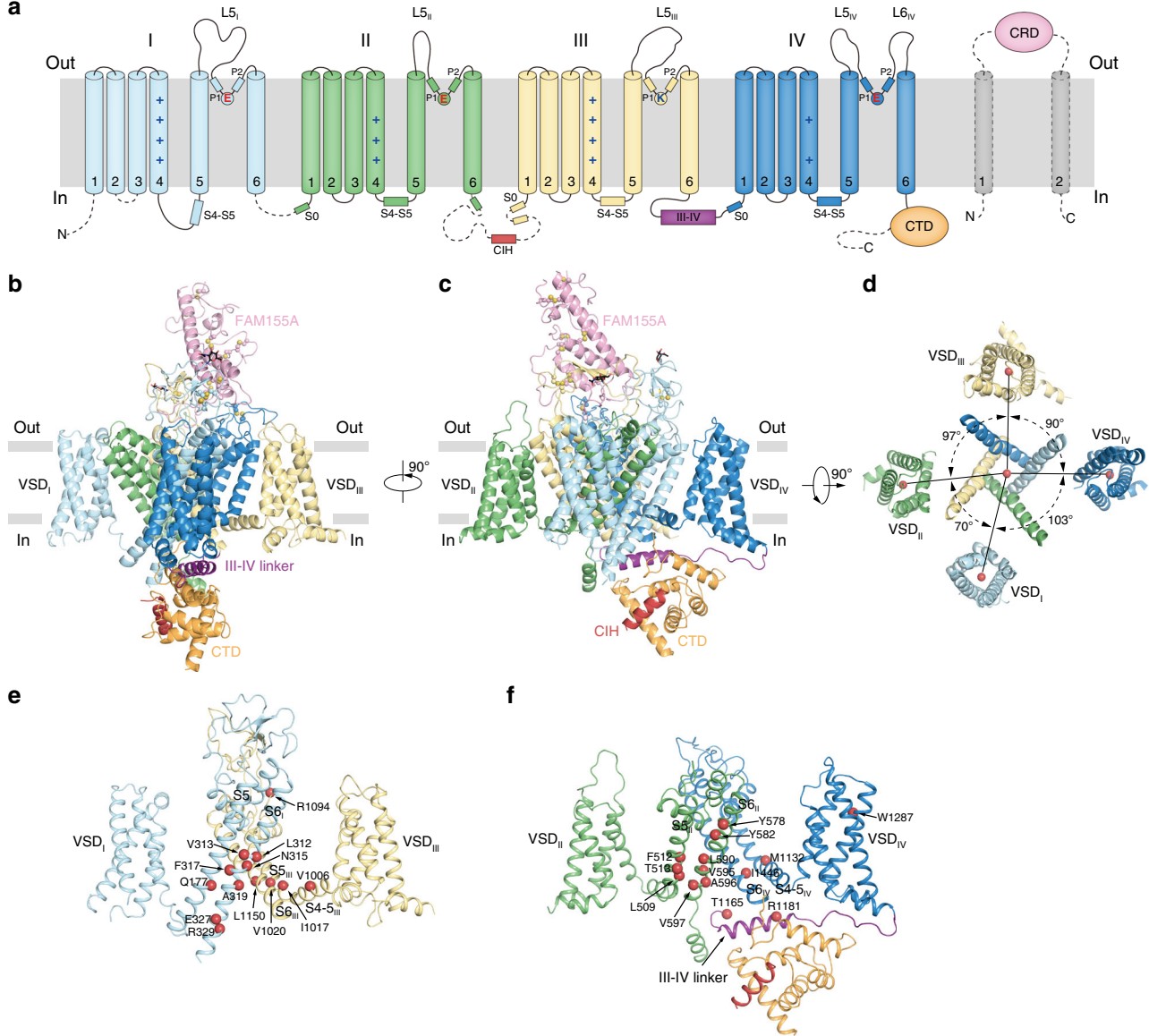

**Fig. 1 Overall structure of NALCN and FAM155A complex. a** Topology of NALCN and FAM155A subunits. Helices are shown as cylinders, unmodeled disordered regions are shown as dashed lines. The phospholipid bilayer is shown as gray layers. CTD C-terminal intracellular domain of NALCN. CIH CTD interacting helix of NALCN. CRD cysteine-rich domain of FAM155A. Plus signs represent positively charged residues on S4 segments. Key residues on the predicted selectivity filter are indicated. **b** Side view of NALCN and FAM155A complex. Sugar moieties are shown as black sticks, side chains of cysteine residues participating in disulfide bond formation are shown as spheres. Sulfur atoms of disulfide bonds were colored in gold and Cα and Cβ atoms were colored the same as each domain. The approximate boundaries of phospholipid bilayer are indicated as gray thick lines. **c** A 90° rotated view compared to **b**. **d** The arrangement of the NALCN transmembrane domain illustrated in the top view. For clarity, only voltage sensors and S6 segments are shown. The angles between adjacent voltage sensors are labeled. Angular measurements were based on the center of mass positions (red spheres) of each domain. **e, f** Structural mapping of disease-related mutations in NALCN. For clarity, only two nonadjacent domains are shown in each panel. The Cα atoms of disease-related residues are shown as red spheres.

The calculated pore profile reveals two constrictions within the transmembrane domain (Fig. 3b, c). The constriction close to the extracellular side is the high field strength (HFS) layer of the selectivity filter[28]. At this layer, the side chains of E280 (domain I), E554 (domain II), and K1115 (domain III) point to the center of the pore (Fig. 3c–f), which is similar to Na$_V$ or Ca$_V$ (Fig. 3g). Unexpectedly, we found the 1387–1390 region of domain IV has two possible alternative conformations, both of which were modeled in the allowed region in the Ramachandran plot validated by Phenix[29] (Supplementary Fig. 7e–h). In one conformation (E1389-I), E1389 makes electrostatic interaction with R1384 on the P1 helix of domain IV and makes hydrogen

bonding with Y287. In the other conformation (E1389-II), E1389 interacts with Y1431 and the main chain of G1388 via hydrogen bonds (Supplementary Fig. 7i). Although the side chain of E1389-II and adjacent D1390 point to the direction of the ion permeation pathway, the side chain of D1390 is above the HFS layer while the E1389-II is below the HFS layer (Fig. 3f, g). This is in agreement with recent findings that the sodium current of NALCN can be blocked by extracellular calcium at physiological concentration and mutations of D1390A has a more profound effect than E1389A on extracellular calcium block[1]. Therefore, there are no side chains on domain IV that is at the corresponding position of the highly negatively charged Glu residue of domain IV in Ca$_V$ at

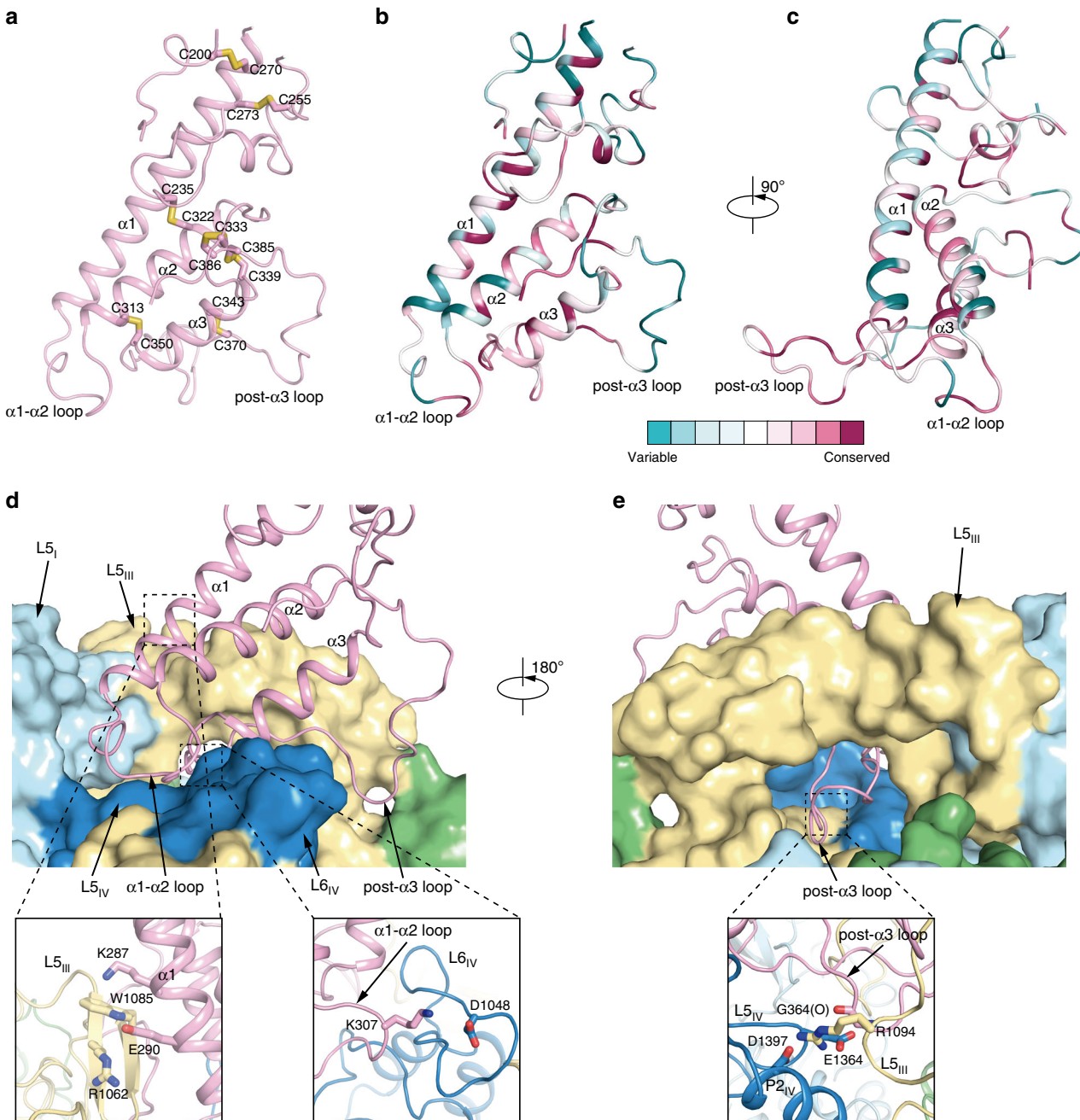

**Fig. 2 Structure of FAM155A CRD and its interactions with NALCN. a** Structure of FAM155A CRD (residues 188–200, 226–262, and 267–389). Disulfide bonds are shown as yellow sticks. **b** Structural mapping of conserved residues of FAM155 family members, generated by ConSurf[50]. **c** A 90° rotated view compared to **b**. **d** Interactions between FAM155A and NALCN. FAM155A is shown as a cartoon. NALCN is shown as a surface. Key interaction regions are boxed and zoomed in the close-up views below. **e** A 180° rotated view compared to **d**.

the HFS layer, such as the E1323 on Ca$_V$1.1 (PDB ID: 5GJV, Fig. 3f). Therefore, the arrangement of charged residues at the HFS layer of NALCN selectivity filter (EEK_, − − + 0) is similar to Na$_V$ (DEKA, − − + 0) but different from Ca$_V$ (EEEE, − − − −) (Fig. 3g), providing a plausible explanation of the sodium selectivity of NALCN. The conformation and structural arrangement of E1389 and D1390 in NALCN reported here were not previously observed in other Na$_V$ or Ca$_V$ channels. To further explore the roles of E1389 and D1390 on the ion permeation, we mutated each of them into alanine and measured the ion selectivity of the mutants. We found mutations of E1389A slightly lowered the $P_{Na}/P_K$ and increased $P_{Na}/P_{Cs}$, while D1390A had no effect on $P_{Na}/P_K$ but markedly increased $P_{Na}/P_{Cs}$ (Fig. 3h),

suggesting although the side chains of both E1389 and D1390 are not on the HFS layer, they may also influence ion permeation through maintaining the local chemical environment. We did not observe strong ion densities inside the pore, indicating sodium ions bind weakly to the pore of NALCN in our sample preparation condition. In comparison with the structure of human NALCN–FAM155A complex (EMD-22203 and PDB ID: 6XIW)[18] (Supplementary Fig. 7a), we found, the extra density of the putatively modified Y287 in 6XIW[18] is contributed by the guanidine group of R1384 in our model, evidenced by the strong connective side-chain density of R1384 in our high-resolution map (Supplementary Fig. 7b–d). We noticed their protein sample was supplemented with 3 mM CaCl$_2$[18], while our sample was not.

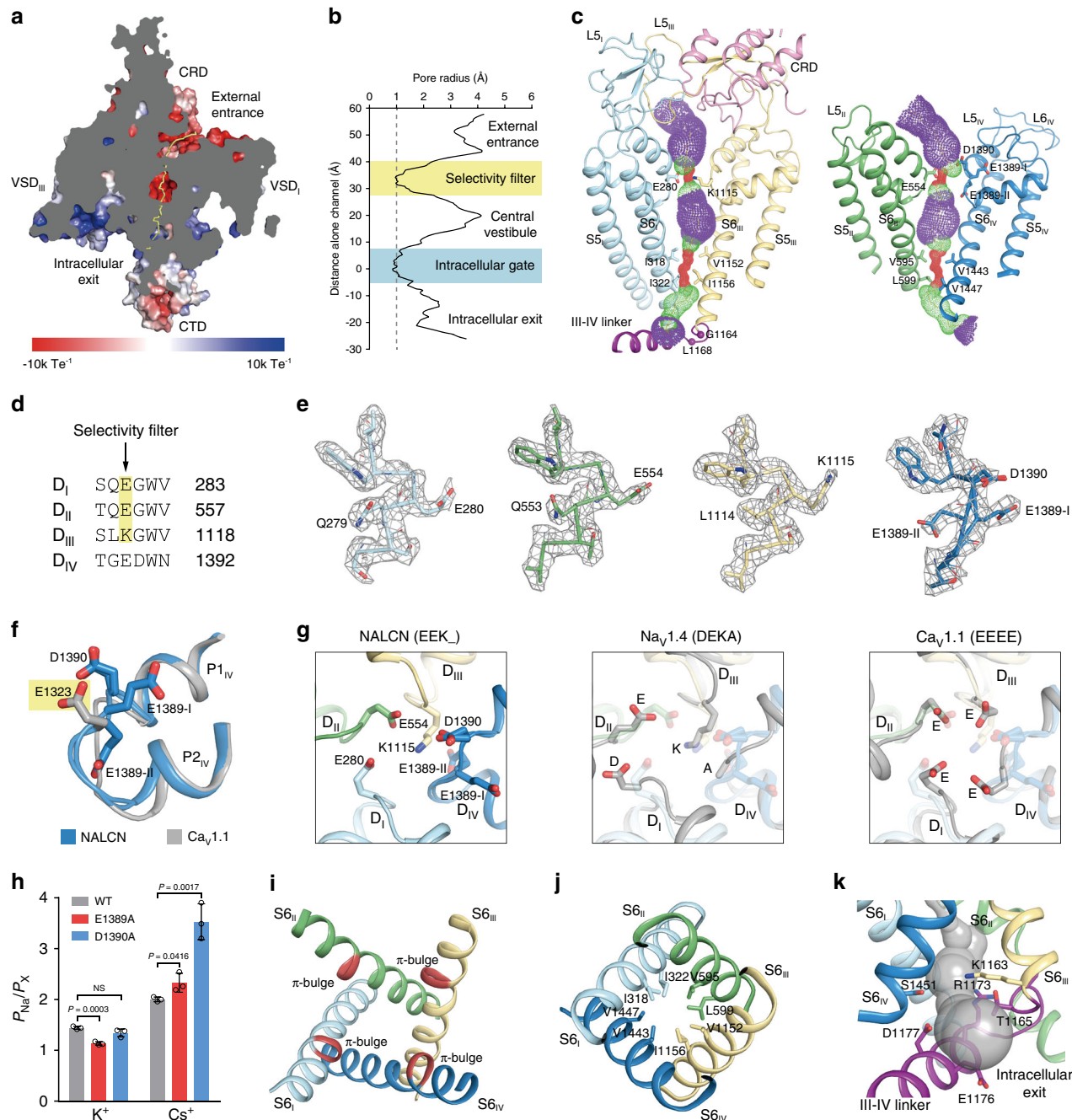

**Fig. 3 Structure of the pore domain of NALCN. a** A cut-open side view of NACLN and FAM155A complex. The surface electrostatic potential is calculated by APBS[51]. The ion permeation pathway is shown as a yellow line. **b** Pore profile of NALCN channel shows two constrictions. **c** The ion permeation pathway of the pore domain, calculated with HOLE[52], is shown as color dots. Purple, green, and red dots represent pore radii >2.8 Å, 1.4–2.8 Å, and <1.4 Å, respectively. **d** Sequence alignment of the residues around the selectivity filter of rat NALCN. Residues that participate in ion permeation are highlighted in yellow. Residues for predicted selectivity filter are indicated with an arrow. **e** Densities of the selectivity filter of each repeat, shown as gray mesh, are contoured at 3.5 σ. **f** Side view of the structural comparison of domain IV between Ca$_V$1.1 (gray) and NALCN (blue). Key residues are shown as sticks. The yellow box indicates the approximate position of the high field strength layer of the selectivity filter. **g** Structural comparisons of the selectivity filter of NALCN, Na$_V$1.4 (PDB: 6AGF), and Ca$_V$1.1 (PDB: 5GJV). Key residues are shown as sticks. **h**, Relative ion permeability ratios ($P_{Na}/P_X$) of NALCN and its mutants. Data are presented as mean ± s.d., $n = 3$ biologically independent cells. Two-tailed unpaired Student's $t$-test was applied, and $P$-values were indicated in the figure. $P < 0.05$ was considered statistically significant. NS no significance. Source data are provided as a Source Data file. **i** Structure of S6 segments. π-bulges are highlighted in red. **j** Top view of the intracellular gate, hydrophobic residues forming the narrowest constrictions are shown as sticks. **k** Side view of the intracellular exit for sodium. The gray surface represents the calculated ion permeation pathway.

Whether these structural differences were due to distinct sample preparation conditions awaits further investigations.

At the level of the central vestibule, the S6 helices of all of the four domains show π-bulges in the middle (Fig. 3i). It is suggested that the α–π helix transition is important for the gating of some channels[30], but its function in NALCN remains unclear. The constriction close to the cytoplasmic side is at the level of bundle crossings which represents the putative intracellular gate of NALCN. The gate is

tightly sealed by several hydrophobic residues on S6 helices (Fig. 3b, c, j), indicating the current structure represents a non-conductive state. Below the gate, the intracellular exit of the ion permeation pathway is blocked by III–IV linker at one side but remains laterally open towards the direction of domain III (Fig. 3k). We found many disease mutations of NALCN accumulate in the pore, around the bundle crossing region, suggesting they might exert their pathological functions by modulating NALCN gating directly (Fig. 1e, f).

**Conformations of degenerated voltage sensors at depolarizing membrane potential.** VS senses changes of membrane potential and drives the gating movement of voltage-gated ion channels. Previous studies showed that the consecutive positive gating charges reside on one face of the S4 helix. On the opposing S2 helix, negative or polar residues are separated by an aromatic residue (Phe or Tyr) and followed by a positively charged residue, forming the E1-X2-F3-E4-R5 structural motif. Together with a negatively charged residue on S3 helix, F3 and E4 on S2 form the so-called charge-transfer center (CTC) at the bottom of VS[31]. The S4 helix of functional VS is in the "down" conformation due to the negative voltage at resting membrane potential, while at depolarizing potential, the S4 helix moves "up" in response to the

changes of electrostatic field[32,33]. In contrast to the canonical voltage-gated ion channels, some of the voltage sensors of NALCN are degenerated due to mutations of key residues in either charge-transfer center on S2 or gating charges on S4. However, NALCN is modulated by voltage[1]. Therefore, the structure of NALCN provides a unique opportunity to explore the conformations of degenerated VS at 0 mV, a depolarizing state.

The voltage sensor of domain I (DI-VS) has a consensus S2 helix. On the S4 helix, positively charged residues of R2, R3, and R5 are conserved, while the middle R4 is replaced by Ile (I149) (Fig. 4a). We observed R2 (R143) interacts with E128 on S3, R3 (R148) interacts with E1 (D75) on S2, and R5 (R152) interacts with E4 (E85) on S2. The Cα of F3 (Y82) is at a level similar to the Cα of R4 (I149) (Fig. 4b). For reference, the Cα of F3 (Y168) in DI-VS of Na$_V$1.4 in depolarizing state (PDB ID: 6AGF) is at a similar level to R3 (R225), suggesting S4 of NALCN DI-VS is even "upper" than Na$_V$1.4 (Supplementary Fig. 9a). The conserved structural features and depolarized conformation of DI-VS suggest it is a functional voltage sensor. Indeed, recent studies showed wild-type NALCN responses to voltage pulse at holding potential of 0 mV but the R146Q + R152Q (R3 + R5) mutant of DI-VS does not[1]. The S4-S5 linker of available Na$_V$ or Ca$_V$ structures

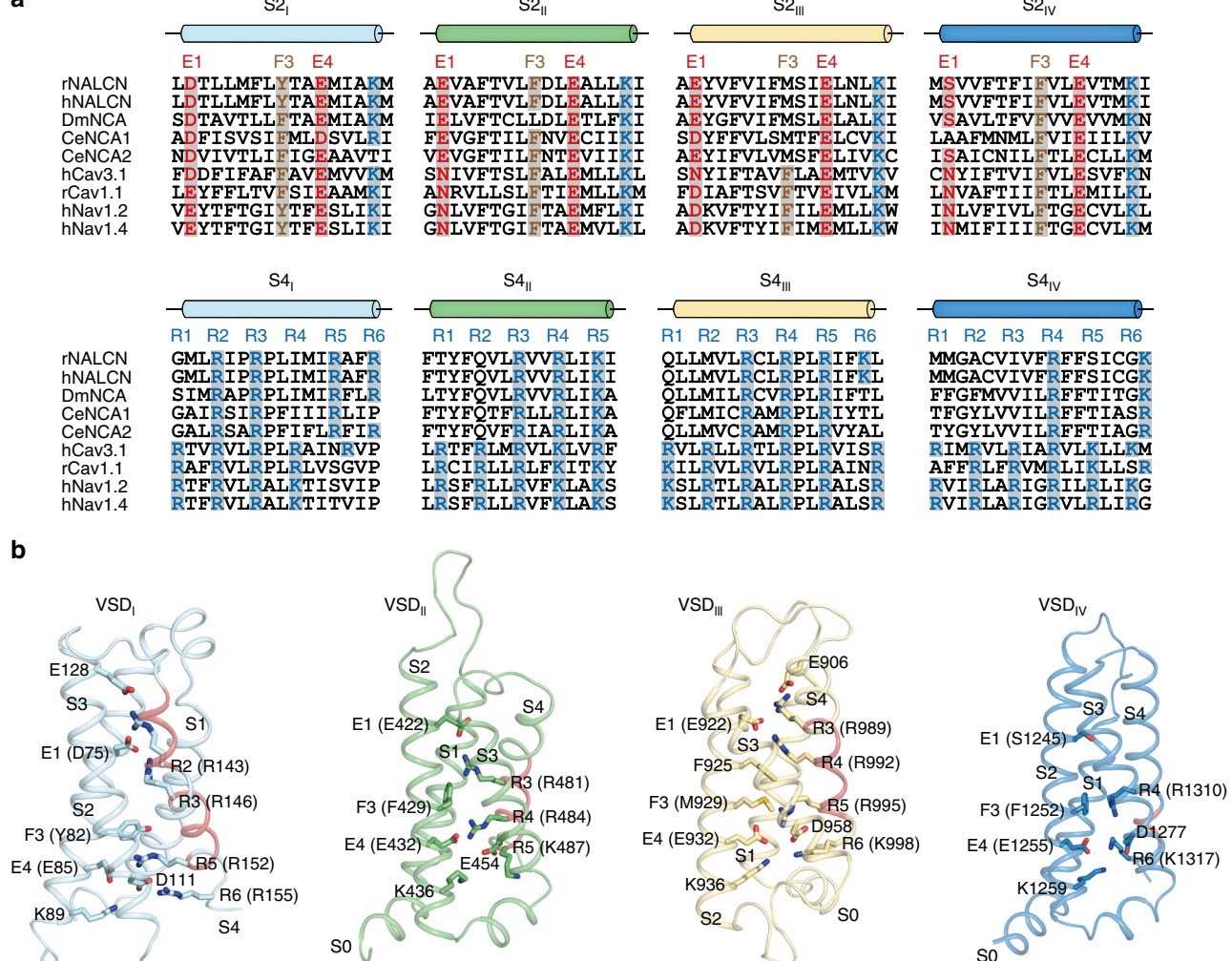

**Fig. 4 Voltage sensors of NALCN at the depolarized state. a** Sequence alignment of S2 and S4 segments. The sequences of rat NALCN, human NALCN, fruit fly NCA, nematode NCA1, NCA2, human Ca$_V$3.1, rabbit Ca$_V$1.1, human Na$_V$1.2, and human Na$_V$1.4 were aligned using PROMALS3D[53]. Residues forming the charge-transfer center are shaded in gray. The conserved negative or polar E1 and E4 on S2 segments, F3 on S2 segments, and positive R1-R6 on S4 segments are highlighted in red, brown, and blue, respectively. **b** Structures of four VSDs in NALCN. Gating charges and charge-transfer center residues are shown as sticks. 3$_{10}$ helices are colored in pink.

adopts a helical structure which is parallel to the membrane plane. However, the N terminus of S4-S5 linker of NALCN DI shows a loop-like structure while its C terminus is fused with S5 to form an exceptionally long S5 helix (Supplementary Fig. 9b). Moreover, the interaction between DI-VS and the pore of domain II is distinct from other VSs (Supplementary Fig. 9c), explaining why the DI-VS is off-axis (Fig. 1d). These structural observations indicate DI-VS of NALCN might convey the voltage signal to the pore in a manner that is distinct from other VSs.

DII-VS has an intact CTC and three consecutive gating charges on S4 (R3-R5) and the Cα of F3 (F429) is at a similar level to Cα of R3 (R481), which is similar to DII-VS of Nav1.4 (PDB ID: 6AGF) (Fig. 4 and Supplementary Fig. 9a). These observations suggest DII-VS is a functional VS. Correlating with this, functional studies showed mutation of R3 (R481) together with one of R4 (R484) or R5 (K487) render NALCN unresponsive to pulse at 0 mV[1].

Although DIII-VS of NALCN has three conserved consecutive gating charges on S4 (R3-R5), it has a defective CTC on S2 and the essential aromatic residue at F3 is replaced by M929 (Fig. 4a). This is consistent with functional studies showing neutralizing mutation of all the gating charges on S4 (R989Q + R992Q + R995Q) has little effect on the voltage modulation of NALCN channel[1]. We observed S4 residues above R3 (R989) adopt α helical structure while residues below R3 form a $3_{10}$ helix. R3 interacts with E906 on S1–S2 linker and E1 (E922). R4 interacts with E1 (E922) and F925. R5 is stabilized by interactions with E4 and D958 on S3. The Cα of F3 (M929) is at the same level as Cα of R5 (R995), which is similar to Nav1.4 (PDB ID: 6AGF) (Fig. 4b and Supplementary Fig. 9a), suggesting DIII-VS of NALCN is in an "up" conformation, although its CTC is defective.

In contrast to the DIII-VS, which has a defective CTC but three conserved gating charges, the DIV-VS of NALCN has conserved CTC but only two gating charges R4 (R1310) and R6 (K1317) on S4. Moreover, the majority of S4 adopts an α helical structure on which the spatial arrangement of gating charges is not optimal for movements upon membrane potential changes, indicating DIV-VS is defective. In agreement with this, mutation of R1310Q has no effect on the voltage sensitivity of NALCN[1]. Notably, it is reported the W1287L mutation on S3 of DIV-VS can leads to the loss of currents and IHRRF[7,34] (Fig. 1f), probably by affecting protein folding or stability.

**Intracellular domains**. At the intracellular side of domain III, $S6_{III}$ helix is connected to the well-ordered III–IV linker via two sharp turns at G1164 and L1168. These structures reroute the intracellular exit for sodium towards the direction of domain III (Fig. 3c). There is an extensive polar interaction network between III–IV linker and the intracellular side of domain I and II, involving E327 and R329 of domain I, E603, K609, and Q613 of domain II, Q1172, and R1173 of III–IV linker. Residues following this turn of III–IV linker fold into the III–IV helix which binds below domain IV. R1174 on this helix interacts with E327 on $S6_I$, while R1181 interacts with the main-chain carbonyl group of S1451, L1452, and Y1454 (Fig. 5a). The gain-of-function R1181Q mutation on this helix causes intellectual disability, episodic and persistent ataxia, arthrogryposis, and hypotonia[20]. Moreover, T1165P mutation can lead to CLIFAHDD syndrome, further emphasizing the important function of III–IV linker (Fig. 1f). The position of III–IV linker in NALCN shares similarity to $Ca_V1.1$ (PDB ID: 5GJV)[35] and $Na_VPaS$ (PDB ID: 5X0M)[36] channels but distinct from $Na_V1.2$ (PDB ID: 6J8E)[37], $Na_V1.4$ (PDB ID: 6AGF)[38], $Na_V1.5$ (PDB ID: 6UZ3)[39], and $Na_V1.7$ (PDB ID: 6J8G)[40] (Supplementary Fig. 10a). NALCN does not have the IFM motif on the III–IV linker, which is involved in the fast inactivation of $Na_V$. This is in accordance with the fact that NALCN does not show fast inactivation during depolarization.

The III–IV helix, together with the intracellular surface of NALCN forms a platform for the binding of CTD. The resolved portion of NALCN CTD consists of five helixes (C1-C5) and forms a calmodulin-like domain that packs below the III–IV linker (Fig. 5b). The NALCN CTD shares structural similarity to those of $Ca_V1.1$ (PDB ID: 5GJV)[35] and $Na_VPaS$ (PDB ID: 5X0M)[36], except the C-terminal C5 helix orients differently (Supplementary Fig. 10b, c). We found a strong extra helical density bound in a groove of CTD, which is cradled by C1–C4 helices. The excellent map density of this helix matches the amino acid sequence of a fragment of the II–III linker (751–765) (Supplementary Fig. 10d and Fig. 5b). Furthermore, GST pull-down results using either protein expressed in HEK293 cells or protein purified from *E. coli* confirmed the interactions between 751–765 fragment and the CTD of NALCN (Supplementary Fig. 10e, f). Therefore, we tentatively named this helix (751–765) as CTD Interacting Helix (CIH). Based on this model, the hydrophobic side chains of the N-terminal I753 and L754 of CIH are inserted into a hydrophobic pocket formed by C1–C4 (Fig. 5c). The positively charged R761, R764, and R765 residues on the C terminal of CIH form a complex interaction nexus with the side chains of D1543 on C4–C5 loop, E1557 and E1560 on C5, and several main-chain carbonyl groups on C4–C5 (Fig. 5c). The interactions between II–III linker and CTD are not observed previously in other $Na_V$ or $Ca_V$ structures. CIH in our model was built as 1589–1602 of

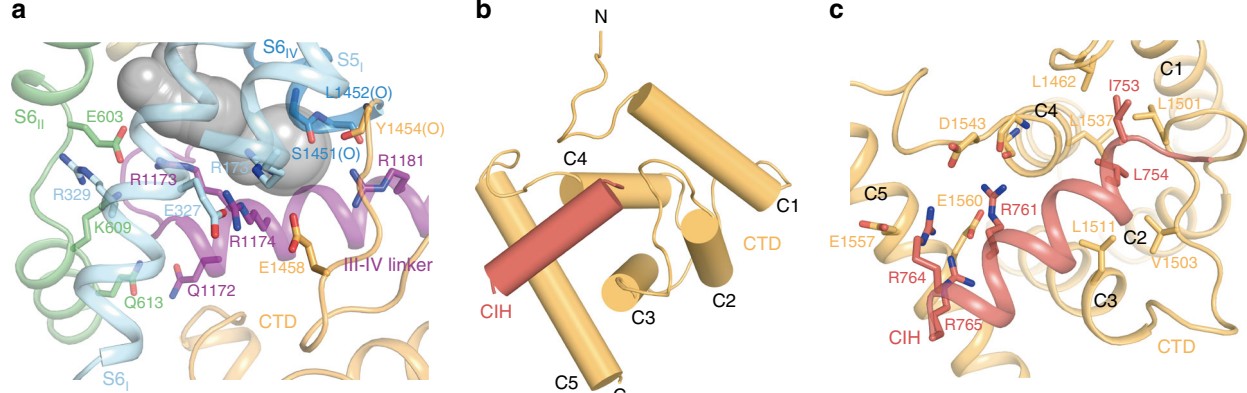

**Fig. 5 Structure of NALCN intracellular domains. a** III–IV linker has extensive polar interactions with CTD and S6 segments in NALCN. The interacting residues are shown as sticks. The gray surface represents the calculated ion permeation pathway. **b** Structure of CTD and CIH of NALCN. Helices are shown as cylinders. **c** Interactions between CTD and CIH of NALCN. The interacting residues are shown as sticks.

NALCN in previously published structure (PDB ID: 6XIW)[18], probably due to the poor local map quality.

## Discussion

The high-resolution structure of the NALCN–FAM155A complex presented here shows the architecture of the NALCN channel core components, reveals the structure of FAM155 CRD, and depicts the detailed inter-molecular interface between NALCN and FAM155A subunits. More importantly, the non-canonical configuration of selectivity filter suggests the mechanism of sodium permeation and calcium block. Furthermore, the asymmetric spatial arrangement of both functional and degenerate voltage sensors provides mechanistic clues for the voltage sensitivity of NALCN. In addition, our structure also provides a template for drug discovery targeting the NALCN channel in related human diseases.

## Methods

**Cell culture.** Sf9 insect cells (ThermoFisher Scientific) were cultured in SIM SF (Sino Biological) at 27 °C. HEK293F suspension cells (ThermoFisher Scientific) were cultured in Freestyle 293 medium (ThermoFisher Scientific) or SMM 293T-I medium (Sino Biological) supplemented with 1% FBS at 37 °C with 6% CO₂ and 70% humidity. HEK293T (ATCC) cells were cultured in Dulbecco's Modified Eagle Medium (DMEM, ThermoFisher Scientific) supplemented with 10% FBS and 1% penicillin-streptomycin (ThermoFisher Scientific) at 37 °C with 5% CO₂. The cell lines were routinely checked to be negative for mycoplasma contamination.

**Electrophysiology.** Constructs containing desired point mutations of rNALCN were generated by Quick Change. HEK293T cells were co-transfected with rNALCN-CGFP, mFAM155A-FLAG, mScarlet-mUNC80, and HA-mUNC79 plasmids (in a modified BacMam expression vectors[41]) at a ratio of 2:1:1:1 using Lipofectamine 3000 (ThermoFisher Scientific) and incubated for 24 h before recording. Patch electrodes were pulled with a horizontal microelectrode puller (P-1000, Sutter Instrument Co, USA) to a resistance of 2–3 MΩ. Whole-cell patch clamps were performed using an Axon-patch 200B amplifier (Axon Instruments, USA), and data were collected with pClamp 10 software (Axon Instruments, USA) and an Axon Digidata 1550B digitizer (Axon Instruments, USA). Pipette solution containing (mM): 10 HEPES (pH 7.2, NaOH), 136 NaCl, 10 NaF, 5 EGTA, 2 Na₂ATP and bath solution containing (mM): 10 HEPES (pH 7.4, NaOH), 150 NaCl, 30 glucose. In some experiments, 10 μM GdCl₃ (Sigma) was added to the bath solution to block the inward current, as indicated. For ion selectivity experiments, the bath solution was exchanged to 10 mM HEPES (pH 7.4, XOH or HCl), 150 mM XCl (X = Na⁺, K⁺, Cs⁺), or NMDG and 30 mM glucose by local perfusion equipment (MPS-2, InBio). The membrane potential was increased from holding potential (0 mV) to +80 mV for 0.5 s to activate the channel and then stepped to various testing potentials (−80 to +80 mV). The peak of tail currents was used to generate $I–V$ curves for the determination of reversal potentials ($E_{rev}$). When Na⁺ in the bath solution, the measured $E_{rev}$ is close to the theoretical $E_{rev}$ (0 mV). To improve the accuracy of our measurements, patches with loose seals (<−100 pA at −80 mV when exposing to NMDG⁺ bath solution) or small currents (<1.5 nA at +80 mV when exposing to Na⁺ bath solution) were discarded. The liquid junction potentials were calculated using pClamp 10 (−4.2 and −4.7 mV for extracellular K⁺ and Cs⁺, respectively) and corrected to calculate the reversal potentials. Signals were acquired at 5 kHz and low-pass filtered at 1 kHz. The ion permeability ratios were calculated with the following equation:

$$P_{Na}/P_X = [X]_o \exp\{[E_{rev}(Na) - E_{rev}(X)]F/RT\}/[Na]_i, \tag{1}$$

where $E_{rev}$, $F$, $R$, and $T$ are the reversal potential, Faraday constant, gas constant, and absolute temperature, respectively. The data were processed with Microsoft Excel and GraphPad Prism 6.

**Protein expression and purification.** Mouse FAM155A was cloned from the cDNA that was reverse transcribed from the total mRNA of the mouse brain. The cDNAs of rat NALCN and mouse FAM155A were cloned into a modified BacMam expression vector with C-terminal GFP-strep and FLAG-tag, respectively[41]. The baculoviruses were produced using the Bac-to-Bac system and amplified in Sf9 cells[42]. For protein expression, HEK293F cells cultured in Freestyle 293 medium at a density of $2.5 \times 10^6$ ml⁻¹ were infected with 6% volume of NALCN P2 virus and 4% volume of FAM155A P2 virus. 10 mM sodium butyrate was added to the culture 12 h post-infection and transferred to a 30 °C incubator for another 48 h before harvesting. Cells were collected by centrifugation at 3990×g (JLA-8.1000, Beckman) for 10 min, and washed with 50 mM Tris (pH 7.5), 150 mM NaCl, 2 μg/ml aprotinin, 2 μg/ml pepstatin, 2 μg/ml leupeptin, flash-frozen, and storage at −80 °C.

For each batch of protein purification, cell pellet corresponding to 0.75 l culture was thawed and extracted with 54 ml lysis buffer containing 50 mM Tris (pH 7.5), 150 mM NaCl, 2 μg/ml aprotinin, 2 μg/ml pepstatin, 2 μg/ml leupeptin, 10% (v/v) glycerol, 1 mM phenylmethanesulfonyl fluoride (PMSF), and 1% (w/v) glyco-diosgenin (GDN, Anatrace) at 4 °C for 80 min. 1 mg/ml iodoacetamide was added during the detergent extraction procedure to reduce non-specific cysteine crosslinking. Then the cell debris was removed by centrifugation at 30,966×g (JA-25.50, Beckman) for 10 min. The supernatant was ultra-centrifuged at 126,100×g (Type 45 Ti, Beckman) for 40 min. The solubilized proteins were loaded onto 3 ml Streptactin Beads 4FF (Smart-Lifesciences) column and washed with 15 ml W buffer, which containing 20 mM Tris (pH 7.5), 150 mM NaCl, 2 μg/ml aprotinin, 2 μg/ml pepstatin, 2 μg/ml leupeptin, 10% glycerol and 0.02% GDN. The column was washed with 50 ml W buffer plus 10 mM MgCl₂ and 2 mM adenosine triphosphate (ATP) to remove contamination of heat shock proteins. Then the column was washed with 20 ml W buffer again to remove residual MgCl₂ and ATP. The target protein was eluted with 12 ml W buffer plus 40 mM Tris (pH 8.0) and 5 mM D-desthiobiotin (IBA). Eluted protein was concentrated using 100-kDa cutoff concentrator (Millipore) and further purified by Superose 6 increase (GE Healthcare) running in a buffer containing 20 mM Tris (pH 7.5), 150 mM NaCl, and 0.02% GDN. Fractions corresponding to monomeric NALCN–FAM155A complex were pooled and concentrated to $A_{280} = 5$.

**Fluorescence-detection size-exclusion chromatography.** 10 μl of purified proteins were injected onto a Superose 6 increase 5/150 column (GE Healthcare), running in a buffer containing 20 mM Tris (pH 7.5), 150 mM NaCl, 0.5 mM $n$-dodecyl β-D-maltoside (DDM, Anatrace), and detected by a fluorescence detector (Shimadzu, excitation 488 nm and emission 520 nm for GFP signal) at room temperature.

**EM sample preparation.** Holey carbon grids (Quantifoil Au 300 mesh, R 0.6/1) were glow-discharged by Solarus advanced plasma system (Gatan) for 120 s using 25% O₂ and 75% Ar. Aliquots of 3 μl concentrated protein sample were applied on glow-discharged grids and the grids were blotted for 2 s before plunged into liquid ethane using Vitrobot Mark IV (ThermoFisher Scientific).

**Cryo-EM data acquisition.** Cryo-grids were firstly screened on a Talos Arctica electron microscope (ThermoFisher Scientific) operating at 200 kV with a K2 Summit direct electron camera (ThermoFisher Scientific). The screened grids were subsequently transferred to a Titan Krios electron microscope (ThermoFisher Scientific) operating at 300 kV with a K2 Summit direct electron camera and a GIF Quantum energy filter set to a slit width of 20 eV. Images were automatically collected using SerialEM in super-resolution mode at a nominal magnification of ×130,000, corresponding to a calibrated super-resolution pixel size of 0.5225 Å with a preset defocus range from −1.5 to −1.8 μm. Each image was acquired as a 7.12 s movie stack of 32 frames with a dose rate of 7.15 e⁻/Å²/s, resulting in a total dose of about 50.9 e⁻/Å².

**Image processing.** The image processing workflow is illustrated in Supplementary Fig. 3. A total of 5,470 super-resolution movie stacks were collected. Motion-correction, two-fold binning to a pixel size of 1.045 Å, and dose weighting were performed using MotionCor2[43]. Contrast transfer function (CTF) parameters were estimated with Gctf[44]. Micrographs with ice or ethane contamination and empty carbon were removed manually. A total of 2,226,376 particles were auto-picked using Gautomatch from 5,428 micrographs. All subsequent classification and reconstruction were performed in Relion 3.1[45] unless otherwise stated. Two rounds of reference-free 2D classification were performed to remove contaminants, yielding 567,397 particles. The particles were subjected to 57 iterations $K = 1$ global search 3D classification with an angular sampling step of 7.5° to determine the initial alignment parameters using the initial model generated from cryoSPARC[46]. For each of the last seven iterations of the global search, a $K = 6$ multi-reference local angular search 3D classification was performed with an angular sampling step of 3.75° and a search range of 30°. The multi-reference models were generated using reconstruction at the last iteration from global search 3D classification low-pass filtered to 8, 15, 25, 35, 45, and 55 Å, respectively. The classes that showed obvious secondary structure features were selected and combined. Duplicated particles were removed, yielding 292,877 particles in total. These particles were subjected to additional three rounds of multi-reference local angular search 3D classification using the same parameters. Each round had 30 iterations, and the particles corresponding to good classes from the last 15 iterations of each round were combined, and duplicated particles were removed, yielding 201,654 particles. These particles were subsequently subjected to local search 3D auto-refinement, which resulted in a map with a resolution of 3.0 Å. In order to further clean up the data set, three rounds of random-phase 3D classification[47] were performed with $K = 2$, an angular sampling step of 1.875°, and a local search range of 10°, using two reference models, which are the model obtained from previous refinement and the phase-randomized model. Phase-randomized models were generated from the model obtained from the previous refinement using randomize software (from the lab of Nikolaus Grigorieff) for phase-randomization beyond 40, 30, and 20 Å for the first, second, and third

round of 3D classification. For random-phase 3D classification, each round had 30 iterations, and the particles corresponding to good class from the last 15 iterations of each round were combined, and duplicated particles were removed, yielding 135,043 particles, resulting in a 2.9 Å map. CTF refinement was then performed with Relion 3.1, the resolution was improved to 2.8 Å. The particles were then re-extracted, re-centered, and re-boxed from 240 pixels to 320 pixels. After 3D auto-refinement, the resolution reached 2.7 Å.

All of the resolution estimations were based on a Fourier shell correlation (FSC) of 0.143 cutoff after correction of the masking effect. B-factor used for map sharpening was automatically determined by the post-processing procedure in Relion 3.1[45]. The local resolution was estimated with Relion 3.1[45].

**Model building**. The initial model of NALCN alone was generated by SWISS-MODEL based on the structure of rabbit $Ca_v1.1$-α1 subunit (PDB 5GJV)[35]. The model was then fitted into the cryo-EM map using Chimera[48] and rebuilt manually using Coot[49]. Model of FAM155A was built manually using Coot[49]. Model refinement was performed using phenix.real_space_refine in PHENIX[29]. Images were produced using Pymol and Chimera[48].

**GST pull-down assay**. HEK293F cells cultured in SMM 293T-I medium were transfected with indicated plasmids (Supplementary Fig. 10e) at a ratio of 1:1 using polyethyleneimine (PEI 25K, Polysciences). 48 h post-transfection, cells were collected and lysed using 20 mM Tris (pH 7.5), 150 mM NaCl, 2 μg/ml aprotinin, 2 μg/ml pepstatin, 2 μg/ml leupeptin, 1 mM PMSF, and 1% GDN at 4 °C for 30 min. After centrifugation at 20,000×g for 10 min, the supernatants were mixed with Glutathione Sepharose 4B (GE Healthcare) and incubated at 4 °C for 1 h. Then the beads were washed with 20 mM Tris (pH 7.5) and 150 mM NaCl six times. Bound proteins were eluted with 50 mM Tris (pH 8.0), 150 mM NaCl, and 10 mM reduced glutathione.

For western blot, proteins were separated with 12% SDS-PAGE and transferred onto polyvinylidene difluoride (PVDF) membranes. Membranes were blocked using 5% nonfat milk in TBST [25 mM Tris (pH 7.4), 137 mM NaCl, 3 mM KCl, and 0.1% Tween-20] for 1 h at room temperature and incubated with primary antibodies [mouse anti-GST (30901ES10; Yeasen Biotechnology), mouse anti-FLAG (M20008M; Abmart), both of antibodies were diluted 5000 times] for 2 h at room temperature. Then membranes were incubated with horseradish-peroxidase (HRP) labeled secondary antibody (31444; ThermoFisher Scientific, the antibody was diluted 10,000 times) for 1 h at room temperature and developed using High-Sig ECL Western Blotting Substrate (Tanon).

For GST pull-down assay using purified proteins from *E. coli* (Supplementary Fig. 10f), CTD was cloned into a pGEX-6P-1 vector, CIH was cloned into pET-21b vector with C-terminal GFP-His tag, GFP was cloned into pET-21b vector with N-terminal His tag and empty pGEX-6P-1 vector was used to express GST. Overexpression of GST-CTD and CIH-GFP was induced in *E. coli* NiCo21 (DE3) with 0.2 mM isopropyl-β-D-thiogalactoside (IPTG) when the cell density reached $OD_{600} = 0.6$ for 18 h at 16 °C, and overexpression of GFP and GST was induced with 1 mM IPTG for 4 h at 37 °C. The cells were collected and sonicated in lysis buffer containing 50 mM Tris (pH 7.5), 150 mM NaCl, 2 μg/ml aprotinin, 2 μg/ml pepstatin, 2 μg/ml leupeptin, and 1 mM PMSF. Then the cell debris was removed by centrifugation at 30,966×g (JA-25.50, Beckman) for 30 min. For purification of GST and GST-CTD, the supernatants were loaded onto Glutathione Sepharose 4B and washed with 20 mM Tris (pH 7.5), 150 mM NaCl and eluted with 50 mM Tris (pH 8.0), 50 mM NaCl and 10 mM reduced glutathione. For purification of GFP and CIH-GFP, the supernatants were loaded onto TALON (Clontech) and washed with W1 buffer (20 mM Tris (pH 7.5), 300 mM NaCl) and W2 buffer (20 mM Tris (pH 7.5), 150 mM NaCl, 10 mM imidazole) and eluted with 250 mM imidazole (pH 8.0) and 50 mM NaCl. All proteins were further purified with an anion-exchange column (HiTrap Q HP, GE Healthcare) and the peak fractions were pooled for GST pull-down assay. The purified proteins were mixed (10 μM for each) as indicated (Supplementary Fig. 10f) at room temperature for 30 min and loaded onto Glutathione Sepharose 4B. The column was washed with 20 mM Tris (pH 7.5) and 150 mM NaCl for 5 times and eluted with 100 mM Tris (pH 8.0), 150 mM NaCl, and 20 mM reduced glutathione. Proteins were separated with 12% SDS-PAGE and visualized by GFP fluorescence (2500B, Tanon) and coomassie blue staining.

## Data availability
Data supporting the findings of this manuscript are available from the corresponding author upon reasonable request. The cryo-EM map of the NALCN–FAM155A complex has been deposited in the EMDB under code EMD-30470. The atomic coordinate of the NALCN–FAM155A complex has been deposited in the PDB with accession code 7CU3. PDB entries (6AGF, 5GJV, 5X0M, 6J8E, 6UZ3, 6J8G, and 6XIW) used in this study were downloaded from Protein Data Bank. Source data are provided with this paper.

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

## Acknowledgements

We thank Prof. Dejian Ren for providing the cDNA of hNALCN, rNALCN, mUNC80, and mUNC79, Dr. Lear Bridget for providing the cDNA of DmUNC79 and DmUNC80, Dr. Ravi Allada for providing the cDNA of DmNCA and DmNLF-1, and Dr. Yuji Kohara for providing the cDNA of CeNCA1. Cryo-EM data collection was supported by the Electron microscopy laboratory and the Cryo-EM platform of Peking University with the assistance of Xuemei Li, Daqi Yu, Xia Pei, Bo Shao, Guopeng Wang, and Zhenxi Guo. Part of structural computation was also performed on the Computing Platform of the Center for Life Science and High-performance Computing Platform of Peking University. We thank the National Center for Protein Sciences at Peking University in Beijing, China for assistance with negative stain EM. This work is supported by grants from the Ministry of Science and Technology of China (National Key R&D Program of China, 2016YFA0502004 to L.C.), National Natural Science Foundation of China (91957201, 31870833, and 31821091 to L.C., 31900859 to J.-X.W.), Beijing Natural Science Foundation (5192009 to L.C.), and Young Thousand Talents Program of China to L.C., and the China Postdoctoral Science Foundation (2016M600856, 2017T100014, 2019M650324, and 2019T120014 to J.-X.W.). J.-X.W. is supported by the Boya Post-doctoral Fellowship of Peking University and the postdoctoral foundation of the Peking-Tsinghua Center for Life Sciences, Peking University (CLS).

## Author contributions

L.C. initiated the project. Y.K. did the electrophysiology recording, purified proteins, and prepared the cryo-EM samples. Y.K. and J.-X.W. collected the cryo-EM data. Y.K. processed the cryo-EM data. Y.K. and L.C. built and refined the atomic model. Y.K. did the GST pull-down assay. All authors contributed to the manuscript preparation.

## Competing interests

The authors declare no competing interests.
