## [Peer Review File · Nature Communications]

REVIEWER COMMENTS

Reviewer #1 (Remarks to the Author):

NALCN is a voltage gated ion channel that is essential in regulating sodium leak current in hippocampal neurons, and mutations found in NALCN are linked to many human diseases. Kang et al., report the cryo-EM structure of rat NALCN/mouse FAM155A complex at 2.7 Å resolution. This NALCN structure presented in this work reprise previous structures determined by other two labs (Kschonsak et. al, Nature 2020; Xie et. al, bioRxiv 2020). Overall, all three structures are in accord with each other. While similar structures have been reported, the structure described here provides several pieces of novel information. Firstly, the author proposes that, residues E-E-K-D1390, rather than E-E-K-E1389, make up the selectivity filter of NALCN. This result contradicts with the findings in other works. Secondly, the author found that part of II-III linker folds as a α -helix and form extensive interacts with CTD. This result was convincingly validated by in vitro pull-down assay. Overall, this cryo-EM work is of high technical quality and advances our knowledge and understanding of NALCN structure, but I think the present manuscript would be strengthened by addressing the following concerns.

Major concerns:

(1) Based on cryo-EM density, the author claims that E1389 adopts two alternative conformations, which might be energetically equivalent. Judging from Figure 3e, the conformational change of E1389 not only occurs in its side chain, but also in its backbone carbonyl. Surprisingly, such big change in E1389 doesn't cause much conformational change of the filter loop that contains E1389, and doesn't alter the conformation of neighboring residues as well. This phenomenon is a bit unusual and not commonly observed in other protein structures. Could author provide an explanation? To convince the audience that this is the case, the author needs to improve the presentation of Figure 3e. For instance, show close up views of the fitting of this filter loop that contains E1389 in different orientations, and highlight the local chemical environment for both conformations of E1389. In addition, the author needs to calculate the phi/psi angles for E1389 and neighboring residues in both alternative models to check whether they are all located in allowed regions in Ramachandran plot. This information should be included in the main text or in a table. Finally, it would be helpful if the author can add a brief discussion on the structural and functional implication of alternative conformations of E1389.

(2) The author proposed that the selectivity filter of NALCN consist of residues E-E-K-D1390, rather than commonly believed E-E-K-E1389. I think this conclusion is speculative, as it is only indirectly supported by another published work that D1390A has more profound effect than E1389A on reducing the calcium mediated channel inhibition. It is possible that the residues involved in the sodium selectivity and calcium mediated inhibition are different. To convincingly prove that EEKD is NALCN's selectivity filter, the author should mutate E1389 and D1390, and compare the effect of these two mutations on the functionality of NALCN with electrophysiology experiment.

(3) As other two labs have published similar NALCN/FAM155A structures (Kschonsak et al., Nature 2020; Xie et al., bioRxiv 2020), the authors should mention all these publications in the introduction, not only in the discussion, and highlight the novel structural information obtained from this work.

Minor points:

(1) It would be useful if the author could prepare a figure to compare the selectivity filter between NALCN, Nav and Cav, and briefly discuss how a EEKD motif at the selectivity filter confers sodium selectivity on NALCN channel.

(2) The author should calculate the FSC between model and map.

(3) NaCl was present in the sample buffer. Is there any additional density in the pore region that could

be assigned as Na? It is possible to see tightly bound ion from a 2.7Å cryo-EM map.

(4) The author employed random-phase 3d classification method to further clean the particle set. This is rather interesting. The original paper reporting this method should be cited.

(<https://doi.org/10.1016/j.cell.2016.05.022>)

(5) In Fig. S1b, the SEC profile of NALCN sample used in this work shows very broad peak. This kind of sample behavior is significantly different from that in other two related works. Could the author provide an explanation? Why was only the sample from fraction 13 used for structural determination? Have the author checked the sample from other fractions?

Reviewer #2 (Remarks to the Author):

In this manuscript, the Kang et al report the Cryo-EM structure of the sodium leak channel NALCN in complex with FAM155A at 2.7 Å. The structure of the NALCN subunit is similar to that of typical voltage-gated sodium and calcium channels, and to the recently reported structure of NALCN by Kschonsak et al, Nature 2020.

The ion conduction through the NALCN channelosome in mammals and invertebrates is implicated in numerous essential physiological processes, including locomotor behaviors, and respiratory rhythms. A number of naturally occurring mutations has been identified in NALCN channelosome and resulted in its malfunction, which is associated with many devastating diseases, including Infantile Neuroaxonal Dystrophy (INAD) or Autosomal Recessive Syndrome with severe hypotonia, speech impairment, and cognitive delay, bipolar disorder, schizophrenia, Alzheimer's disease, autism, epilepsy, alcoholism, cardiac diseases and cancer. Therefore, an increased understanding of the NALCN structure-functional relationship is of general interest, and structures of NALCN in complex with FAM155 are in good agreement with the previous structure presented by Kschonsak et al in Nature. However, it is great to observe a similar finding from independent groups and provide basic grounding for a functional researcher for the future interrogation of disease mutants.

In this study, authors observed that the cysteine repeat domain (CRD) of the FAM155A complex is resolved and it sits on the top of the NALCN channel. The CRD consists of three alpha-helices that are connected by loop segments. The C-terminal end of the $\alpha 1$ helix, $\alpha 1$ - $\alpha 2$ loop, and the post $\alpha 3$ loop interact with the loops connecting the pore-forming transmembrane segments S5 and S6 of domains I, III and IV. The authors have properly explained the details of the amino acids involved in electrostatic and cation- π interactions responsible for this interaction of CRD with NALCN. In this closed conformation structure of NALCN-FAM155A, the shape of the pore is markedly different from that observed in typical Nav and Cav. The intracellular exit for NALCN is interrupted by a III-IV linker which seems to direct the pathway for the flow of Na⁺ ions towards domain III as shown by HOLE. Other important features described include the participation of D1390 instead of E1389 in the formation of the selectivity filter, the hydrophobic residue of the lower S6 segment forming the gate, and the π -helix bulges occurring in the middle of the S6 segment.

The authors have also sufficiently explained the reason for the voltage-sensitive nature of the voltage sensors (VS) of domain I and domain II. They have shown the amino acid residues involved in forming the charge transfer center (CTC) and gating charges in the S4 segment (which confer voltage sensitivity) are conserved in DI and DII but not in DIII and DIV. This is in perfect agreement with an earlier investigation by H.C. Chua et al., 2019 who showed the same with functional studies of mutations in S4 segments. Based on the conserved structural motif formed by the CTC and arginine residues in the S4 segment, the upward confirmation of S4 segments relative to the S2 and S3 segments agrees with the depolarized state structure of the ion channel which the authors claim. This article is important for the NALCN channel community because it provides a detailed structural analysis for a better understanding of its roles in diseases and would immensely benefit the scientific

community. However, I feel that the current article could be a little bit improved if a few more queries and some questions need to be addressed before accepted for publication.

1. Lipids play an important role in the modulation of channel activity. The resolution of resulting structures is quite high and therefore it is surprising that authors did not report any putative lipid-binding sites. Are there any lipid-like density bound to the channels, if yes, this should be reported in the paper with the proper figure and what are authors thought on these putative lipids.

2. Authors are also suggested to compare different domains of their structures with the recently published structure of NALCN-FAM155 complex Kschonsak et al, Nature 2020, and how their structure looks similar or different to previously reported structures.

3. In methods section, RCF should be reported instead of RPM because RPM is a highly variable parameters.

Additionally, I have the following issues with the text and figures.

List of typographical errors:

- There are errors where "NALCN" is mis-spelt as "NLACN". The authors are requested to check the following lines and correct the errors. Lines: 55,86, 236, 171, 189.
- In the manuscript, the terms "UNC-79" and "UNC79" are interchangeably used. So are the terms "UNC-80" and "UNC80". Though both forms are correct, I would suggest the authors follow just one form – preferably UNC79/UNC80 – for the sake of uniformity.

List of sentence revisions.

- In lines 96 and 97, I would suggest re-writing the following sentence from:

"The structure of NALCN allowed us to locate all of the reported missense mutations identified in human diseases and model animals" seems confusing and authors are suggested to modify as to either:

"The structure of NALCN allowed us to locate all of the reported missense mutations identified in the NALCN subunit that cause disease in humans and model organisms".

Or

"The structure of NALCN allowed us to locate all of the reported missense mutations identified in the pore-forming subunit of the NALCN channel complex that causes disease in humans and model organisms".

The authors are requested to make this change as the original phrase "...all of the reported missense mutations identified in human diseases and model animals" implies even those mutations reported in other subunits – FAM155A, UNC79, and UNC80 which are known to cause disease. In lines 115 and 116, the authors themselves have mentioned a mutation in FAM155A – R1094Q – that causes disease, which is obviously not present in the NALCN subunit.

- In lines 197, I would suggest re-writing the following sentence from:

"At the intracellular side of domain III, the well-ordered III-IV linker makes two sharp turns at G1164 and I1168 and reroutes the intracellular exit for sodium towards the direction of domain III"

to:

"At the intracellular side of domain III, the S6III helix makes two sharp turns at G1164 and I1168, which is followed by a well ordered III-IV linker that re-routes the intracellular exit for sodium towards the direction of domain III"

The authors are suggested to make the above change as the original sentence conveys the different meanings and confusing – the two sharp turns at G1164 and I1168 form after the III-IV linker not before.

- In line 132, the word "D1389A" should be changed to "E1389A". In the paper by Chua et al 2019,

describing the pharmacological characterization of the NALCN channel complex, there is no mention of D1389A substitution. It has been shown that D1390A substitution is more sensitive to Ca²⁺ block than the E1389A mutation.

The authors are suggested to rectify this error.

List of missing labels in the diagrams.

- In lines 125 and 126, the authors mention that E1389-I in up confirmation making an electrostatic interaction with R1384 on P1 helix of domain IV. (Fig. 3c, f). The R1384 is not labelled in Fig. 3c,f.

The authors are requested to label R1384 in Fig. 3c,f if possible.

- In lines 197, 198 and 199, the authors report two sharp turns occurring at G1164 and I1168 Fig 3c.

The authors are requested to label I1168 either in diagrams Fig 3c or Fig 3i and indicate it appropriately in the text.

- In lines 205, 206, and 207 the authors mention about T1165P.

The authors are requested to indicate either (Fig. 1f) or (Fig 3i) at the end of the sentence "Moreover, T1165P mutation can...III-IV linker."

List of clarifications.

- In Fig 1b, the disulfide bonds are shown as golden spheres. In the same diagram, there are also spheres colored cyan, blue, pink, and yellow ochre.

Can the authors please explain what these spheres of additional colors denote?

- In lines 202 and 203 the authors mention that R1174 interacts with E1458 on the CTD (Fig. 5a). However, from this figure, it looks like R1174 is closer to E327 of domain I (than E1458) seemingly stabilizing the III-IV linker helix to block the pore of the NALCN in this closed conformation.
 - o Is this true? If so, then, do the authors observe E1458 interacting with other positive or polar amino acids in the vicinity?
 - o If this is not true, do the authors observe E327 interacting with other positive or polar amino acid in vicinity?
- In line 219, the performed a GST pull-down assay to confirm the interaction between CTD and CIH (Fig S8e). Here they have used two different antibodies – anti-GST and anti-FLAG.

If the authors had expressed both CTD and CIH with the FLAG tag, they could have performed western blot with just the anti-FLAG antibody. If CTD and CIH interact, there would be two separate bands in the elute fraction, if they do not interact only one band would be seen in elute fraction. Did the authors explore this strategy of GST pull-down? Because this strategy is similar to the one in the paper by H.C. Chua et al.,2019 where UNC79, UNC80, and FAM155A were shown to interact with NALCN.

Reviewer #3 (Remarks to the Author):

Kang et al. determined the high resolution cryo-EM structure of NALCN in complex with FAM155A. The structure reveals detailed interactions between NALCN and extracellular cysteine-rich domain of FAM155A. Their study demonstrate that the asymmetric arrangement of two functional voltage

sensors confers the modulation by membrane potential. The structure of NALCN allowed them to locate all of the reported missense mutations identified in human diseases and model animals. While this study is solid and novel, there are few questions which need to be addressed before publication.

1) There is a recent structure of the same complex in nanodisc by Kschonsak et al as mentioned by author as well. Comparison with that structure is highly recommended especially when they talk about the selectivity filter.

2) The structure is nicely modelled but the fitting at the region of 302-305 and 233-237 need to be improved.

3) They mentioned that the VS domains are highly asymmetric. I recommend to measure the buried area at the interfaces and discuss.

4) The FAM155A is attached to the vicinity of ion entry pathway. Could author shed light on how FAM155A enhance the activity of the channel.

5) The authors have modeled lipids in the structure but they have not discussed about the possible lipid protein interactions and importance of lipids in the modulation of channel structure and function.

6) Intracellular ion exit pathway is blocked in one domain whereas laterally open in another domain. Please calculate the radius of the cavity and show whether it can allow ion to pass through.

7) In figure S2e, show unmasked plot as well.

8) In table S1 include molprobity percentiles, initial model used, map resolution unmasked.

Point-to-point response to reviewer comments

We are grateful to the constructive suggestions from reviewers. We have substantially revised our manuscript, providing new data, revised figures, text and method sections. Please find the point-to-point responses below.

Reviewer #1 (Remarks to the Author):

Major concerns:

(1) Based on cryo-EM density, the author claims that E1389 adopts two alternative conformations, which might be energetically equivalent. Judging from Figure 3e, the conformational change of E1389 not only occurs in its side chain, but also in its backbone carbonyl. Surprisingly, such big change in E1389 doesn't cause much conformational change of the filter loop that contains E1389, and doesn't alter the conformation of neighboring residues as well. This phenomenon is a bit unusual and not commonly observed in other protein structures. Could author provide an explanation? To convince the audience that this is the case, the author needs to improve the presentation of Figure 3e. For instance, show close up views of the fitting of this filter loop that contains E1389 in different orientations, and highlight the local chemical environment for both conformations of E1389. In addition, the author needs to calculate the phi/psi angles for E1389 and neighboring residues in both alternative models to check whether they are all located in allowed regions in Ramachandran plot. This information should be included in the main text or in a table.

Response: In our model, a range of residues (1387-1390) have been modeled to have two alternative conformations, with side chain of E1389 having most dramatic structural differences. The fitting of densities and main chain phi/psi distributions of 1387-1390 on Ramachandran plot are provided in Supplementary Fig. 7e-h and indicated in revised text in line 132-135.

Finally, it would be helpful if the author can add a brief discussion on the structural and functional implication of alternative conformations of E1389.

Response: We agree with reviewer that alternative conformation in ion channel selectivity filter is unprecedented. However, the detailed examination of their biological functions and mechanistic implications awaits extensive studies in the future. Therefore, we prefer to not to speculate too much in the manuscript at the moment.

(2) The author proposed that the selectivity filter of NALCN consist of residues E-E-K-D1390, rather than commonly believed E-E-K-E1389. I think this conclusion is speculative, as it is only indirectly supported by another published work that D1390A has more profound effect than E1389A on reducing the calcium mediated channel inhibition. It is possible that the residues involved in the sodium selectivity and calcium mediated inhibition are different. To convincingly prove that EEKD is NALCN's selectivity filter, the author should mutate E1389 and D1390, and compare the effect of these two mutations on the functionality of NALCN with electrophysiology experiment.

Response: We provided detailed structural comparison between NALCN, Nav and Cav in Fig. 3f-g. In addition, we mutated E1389 and D1390 to alanine individually and measured the ion selectivity of mutants in Fig. 3h and Supplementary Fig. 8. Moreover, we reorganized our structural description and discussion for better illustration of the position and conformations of E1389 and D1390 in line 129-155. We found although the side chains of both E1389 and D1390 are not on the HFS layer, they influence ion permeation probably through maintaining local chemical environment.

(3) As other two labs have published similar NALCN/FAM155A structures (Kschonsak et al., Nature 2020; Xie et al., bioRxiv 2020), the authors should mention all these publications in the introduction, not only in the discussion, and highlight the novel structural information obtained from this work.

Response: These two papers were described and cited in line 67-70.

Minor points:

(1) It would be useful if the author could prepare a figure to compare the selectivity filter between NALCN, Nav and Cav, and briefly discuss how a EEKD motif at the selectivity filter confers sodium selectivity on NALCN channel.

Response: We have provided structural comparison in Fig. 3f-g and related discussion in line 129-148. We found the charge arrangement of the HFS layer is similar to Nav, which explains the Na selectivity.

(2) The author should calculate the FSC between model and map.

Response: We have provided FSC between model and map in Supplementary Fig. 2f

(3) NaCl was present in the sample buffer. Is there any additional density in the pore region that could be assigned as Na? It is possible to see tightly bound ion from a 2.7Å cryo-EM map.

Response: We didn't observe strong density for sodium map. This claim has been included in line 155-156.

(4) The author employed random-phase 3d classification method to further clean the particle set. This is rather interesting. The original paper reporting this method should be cited.

(<https://doi.org/10.1016/j.cell.2016.05.022>)

Response: We have cited this paper in revised manuscript.

(5) In Fig. S1b, the SEC profile of NALCN sample used in this work shows very broad peak. This kind of sample behavior is significantly different from that in other two related works. Could the author provide an explanation? Why was only the sample from fraction 13 used for structural determination? Have the author checked the sample from other fractions?

Response: We have provided the FSEC traces of each individual fractions from SEC in Supplementary Fig. 1d. We found the concentrating procedure prior to SEC lead to the partial aggregation of protein. Therefore, the early fractions of SEC tend to show higher molecular weight, probably due to the aggregation induced by concentration. Therefore we only combined

fractions that are close to the monomeric NALCN-FAM155 complex, indicated by the FSEC peak position of eluate from affinity column.

Reviewer #2 (Remarks to the Author):

However, I feel that the current article could be a little bit improved if a few more queries and some questions need to be addressed before accepted for publication.

1. Lipids play an important role in the modulation of channel activity. The resolution of resulting structures is quite high and therefore it is surprising that authors did not report any putative lipid-binding sites. Are there any lipid-like density bound to the channels, if yes, this should be reported in the paper with the proper figure and what are authors thought on these putative lipids.

Response: We have provided putative lipid densities in Supplementary Fig. 4b and related description in line 90-91. However, we cannot explicitly determine the identity of each lipids, so we did not speculate too much on their functions at the moment.

2. Authors are also suggested to compare different domains of their structures with the recently published structure of NALCN-FAM155 complex Kschonsak et al, Nature 2020, and how their structure looks similar or different to previously reported structures.

Response: We have supplemented structure comparison in Supplementary Fig. 7a and line 102-105.

3. In methods section, RCF should be reported instead of RPM because RPM is a highly variable parameters.

Response: We have replace RPM with RCF in methods section.

Additionally, I have the following issues with the text and figures.

List of typographical errors:

- There are errors where “NALCN” is mis-spelt as “NLACN”. The authors are requested to check the following lines and correct the errors. Lines: 55,86, 236, 171, 189.

Response: We have corrected these typo.

- In the manuscript, the terms “UNC-79” and “UNC79” are interchangeably used. So are the terms “UNC-80” and “UNC80”. Though both forms are correct, I would suggest the authors follow just one form – preferably UNC79/UNC80 – for the sake of uniformity.

Response: We have corrected these typo.

List of sentence revisions.

- In lines 96 and 97, I would suggest re-writing the following sentence from:

“The structure of NALCN allowed us to locate all of the reported missense mutations identified in human diseases and model animals” seems confusing and authors are suggested to modify as to either:

“The structure of NALCN allowed us to locate all of the reported missense mutations identified in the NALCN subunit that cause disease in humans and model organisms”.

Or

“The structure of NALCN allowed us to locate all of the reported missense mutations identified in the pore-forming subunit of the NALCN channel complex that causes disease in humans and model organisms”.

Response: We have revised this sentence in line 100-102.

The authors are requested to make this change as the original phrase “...all of the reported missense mutations identified in human diseases and model animals” implies even those mutations reported in other subunits – FAM155A, UNC79, and UNC80 which are known to cause disease. In lines 115 and 116, the authors themselves have mentioned a mutation in FAM155A – R1094Q – that causes disease, which is obviously not present in the NALCN subunit.

Response: We have revised this sentence in line 100-102 and 123-125.

- In lines 197, I would suggest re-writing the following sentence from:

“At the intracellular side of domain III, the well-ordered III-IV linker makes two sharp turns at G1164 and I1168 and reroutes the intracellular exit for sodium towards the direction of domain III”

to:

“At the intracellular side of domain III, the S6III helix makes two sharp turns at G1164 and I1168, which is followed by a well ordered III-IV linker that re-routes the intracellular exit for sodium towards the direction of domain III”

The authors are suggested to make the above change as the original sentence conveys the different meanings and confusing – the two sharp turns at G1164 and I1168 form after the III-IV linker not before.

Response: We have revised this sentence in line 229-231.

- In line 132, the word “D1389A” should be changed to “E1389A”. In the paper by Chua et al 2019, describing the pharmacological characterization of the NALCN channel complex, there is no mention of D1389A substitution. It has been shown that D1390A substitution is more sensitive to Ca²⁺ block than the E1389A mutation.

Response: We have revised this sentence in line 140-142.

- In lines 125 and 126, the authors mention that E1389-I in up confirmation making an electrostatic interaction with R1384 on P1 helix of domain IV. (Fig. 3c, f). The R1384 is not labelled in Fig. 3c,f. The authors are requested to label R1384 in Fig. 3c,f if possible.

Response: We have provided Supplementary Fig. 7i for better illustration.

- In lines 197, 198 and 199, the authors report two sharp turns occurring at G1164 and I1168 Fig 3c. The authors are requested to label I1168 either in diagrams Fig 3c or Fig 3i and indicate it appropriately in the text.

Response: We have provided revised Fig. 3c to label G1164 and I1168

- In lines 205, 206, and 207 the authors mention about T1165P. The authors are requested to indicate either (Fig. 1f) or (Fig 3i) at the end of the sentence “Moreover, T1165P mutation can...III-IV linker.”

Response: We have cited Fig. 1f at the appropriate position.

List of clarifications.

- In Fig 1b, the disulfide bonds are shown as golden spheres. In the same diagram, there are also spheres colored cyan, blue, pink, and yellow ochre.

Can the authors please explain what these spheres of additional colors denote?

Response: We have provided revised Fig. 1b legend as: "Side chains of cysteine residues participating disulfide bonds are shown as spheres. Sulfur atoms of disulfide bonds were colored in gold and C α and C β atoms were colored the as each domains. "

- In lines 202 and 203 the authors mention that R1174 interacts with E1458 on the CTD (Fig. 5a). However, from this figure, it looks like R1174 is closer to E327 of domain I (than E1458) seemingly stabilizing the III-IV linker helix to block the pore of the NALCN in this closed conformation. o Is this true? If so, then, do the authors observe E1458 interacting with other positive or polar amino acids in the vicinity? o If this is not true, do the authors observe E327 interacting with other positive or polar amino acid in vicinity?

Response: We clarified the interaction residues of R1174 in line 232-233: "R1174 on this helix interacts with E327 on S6_i, and E1458 on CTD interacts with R173 on S5_i."

- In line 219, the performed a GST pull-down assay to confirm the interaction between CTD and CIH (Fig S8e). Here they have used two different antibodies – anti-GST and anti-FLAG. If the authors had expressed both CTD and CIH with the FLAG tag, they could have performed western blot with just the anti-FLAG antibody. If CTD and CIH interact, there would be two separate bands in the elute fraction, if they do not interact only one band would be seen in elute fraction. Did the authors explore this strategy of GST pull-down? Because this strategy is similar to the one in the paper by H.C. Chua et al.,2019 where UNC79, UNC80, and FAM155A were shown to interact with NALCN.

Response: We were worried that truncated protein fragments might be subjected to proteolysis and several bands of the same protein but at different molecular weights could be observed. This

might complicate the interpretation of results when tagging both interaction partners using same tag. Therefore, we chose to tag different subunits with different tags.

Reviewer #3 (Remarks to the Author):

1) There is a recent structure of the same complex in nanodisc by Kschonsak et al as mentioned by author as well. Comparison with that structure is highly recommended especially when they talk about the selectivity filter.

Response: We provided detailed structure comparison in line 102-105 and Supplementary Fig. 7a.

2) The structure is nicely modelled but the fitting at the region of 302-305 and 233-237 need to be improved.

Response: We improved the fitting of model and the statistics of revised model is now provided in Supplementary Table 1.

3) They mentioned that the VS domains are highly asymmetric. I recommend to measure the buried area at the interfaces and discuss.

Response: We calculated the buried area between VS and pore. They are 1393\AA^2 , 1761\AA^2 , 1658\AA^2 , and 1470\AA^2 for DI-VS, DII-VS, DIII-VS and DIV-VS, respectively. We have also provided structural comparison showing DI-VS interacts with pore differently from other VS in revised Supplementary Fig. 9c. This explained why VS arrangement is highly asymmetric.

4) The FAM155A is attached to the vicinity of ion entry pathway. Could author shed light on how FAM155A enhance the activity of the channel.

Response: Because the interaction between FAM155 and NALCN involves an extensive interface which might be important for protein stability of NALCN, we speculate FAM155 assist the folding of NALCN. But we don't have any direct evidence at the moment.

5) The authors have modeled lipids in the structure but they have not discussed about the possible lipid protein interactions and importance of lipids in the modulation of channel structure and function.

Response: We have provided putative lipid densities in Supplementary Fig. 4b and related description in line 90-91. However, we cannot explicitly determine the identity of each lipids, so we did not speculate too much on their functions at the moment.

6) Intracellular ion exit pathway is blocked in one domain whereas laterally open in another domain. Please calculate the radius of the cavity and show whether it can allow ion to pass through.

Response: We have provided revised Fig. 3b and c to show the intracellular exit might allow sodium ion to permeate.

7) In figure S2e, show unmasked plot as well.

Response: We have provided unmasked plot in revised Supplementary Fig. 2e.

8) In table S1 include molprobity percentiles, initial model used, map resolution unmasked.

Response: We have provided these numbers in Supplementary Table 1.

REVIEWERS' COMMENTS

Reviewer #1 (Remarks to the Author):

The authors have satisfactorily addressed all of the points that were raised in my initial review by adding new mutagenesis data and modifying presentation. The manuscript has been improved substantially. The modelling of the alternative conformations of E1389 looks much more convincing with providing the Ramachandran plots. I don't have any concern now. I strongly support its publication at Nature Communications.

Point-to-point response to reviewer comments

Reviewer #1 (Remarks to the Author):

The authors have satisfactorily addressed all of the points that were raised in my initial review by adding new mutagenesis data and modifying presentation. The manuscript has been improved substantially. The modelling of the alternative conformations of E1389 looks much more convincing with providing the Ramachandran plots. I don't have any concern now. I strongly support its publication at Nature Communications.

Response: We are grateful to the constructive suggestions from reviewers.